# Pseudo-Tetrahedral Copper(I) Symmetrical Formamidine Dithiocarbamate-Phosphine Complexes: Antibacterial, Antioxidant and Pharmacokinetics Studies

Segun D. Oladipo [1,2] and Bernard Omondi [1,*]

1   School of Chemistry and Physics, Westville Campus, University of Kwazulu-Natal, Private Bag X54001, Durban 4000, South Africa; segun.oladipo@oouagoiwoye.edu.ng

2   Department of Chemical Sciences, Olabisi Onabanjo University, P.M.B 2002, Ago-Iwoye 120107, Nigeria

*   Correspondence: owaga@ukzn.ac.za; Tel.: +27-31-260-8127

**Abstract:** Three copper(I) dithiocarbamate–phosphine complexes of the general formula $Cu(PPh_3)_2L$ were synthesized by metathesis reactions of the potassium salt of the dithiocarbamate ligand **L** and the precursor complex $Cu(PPh_3)_2NO_3$ in an equimolar ratio. **L** represents N,N′-bis(2,6-dimethylphenyl)formamidine dithiocarbamate **L1** in complex **1**, N,N′-bis(2,6-disopropylphenyl) formamidine dithiocarbamate **L2** in complex **2,** and N,N′-dimesitylformamidine dithiocarbamate **L3** in complex **3**. The single-crystal X-ray structure revealed the coordination of the copper atom to two sulfur atoms of the dithiocarbamates, as well as two phosphorus atoms of the $PPh_3$ units, which resulted in distorted tetrahedral geometries. The calculated $\tau_4$ (tau factor) values for **1**, **2** and **3** were 0.82, 0.81 and 0.85, respectively, confirming the pseudo-tetrahedral geometry proposed. Complexes **1–3** showed remarkable luminescent properties in $CH_2Cl_2$ at room temperature. All three complexes showed moderate-to-low antibacterial potential against Gram-negative bacteria, while none of the complexes were active against Gram-positive bacteria. The DPPH assay studies showed that complex **2** had the lowest $IC_{50}$ ($4.99 \times 10^{-3}$ mM),and had higher DPPH free radical scavenging ability than **1** and **3**. The pharmacological estimations of **1–3** showed that all of the complexes showed minimal violation of Lipinski's rule.

**Keywords:** copper(I); dithiocarbamates; photoluminescence; antibacterial; DPPH assay

## 1. Introduction

Antimicrobial resistance is a fundamental predicament that has been in existence since the advent of the antibiotic period [1]. In the last two decades, bacteria resisting antibiotic drugs have become a global concern for the healthcare sector [2]; therefore, a holistic approach is required to abate this concern. Previous studies have shown that at least one mechanism of resistance is present in the general environment for a recently discovered natural product with antibiotic potential, and this extremely accelerates emergence of antibiotic resistance [3]. The misuse and overuse of commercially available antibiotic drugs is also one of the causes of antimicrobial resistance [4]. Due to the acquired mechanism of antimicrobial resistance along with the presence of multiple intrinsic resistances to several antibiotics, it is now difficult to control the spread of deadly microbes in healthcare settings [5]. One of the possible techniques to overcome antibiotic resistance is to develop novel synthetic drugs of which the chemical structures do not occur naturally [6], hence the increase in research related to the synthesis of novel antibiotics that can outsmart bacteria which are resisting available drugs.

Similarly, *havoc* caused by free radicals in the body system has been reported in literature [7,8]. Though in small quantity, i.e., their generation during normal metabolism in the body system, they are crucial to regular physiological function. However, in surplus, they harm enzymes, proteins and DNA, leading to a high risk of deadly diseases such as

diabetes, Alzheimer's, and Parkinson's disease, among others [7]. For this reason, studies on the development of natural and synthetic antioxidants to counterbalance excess free radicals produced in our body system is on the increase.

Medical properties such as antimicrobial [9], antioxidant [5], anticancer [10], and antidiabetes [11] activities of copper complexes have been reported. It is important to know that, among copper's eminent oxidation states, Cu(I) has remarkable antibacterial potential [12]. Cu(I) is readily available in human blood, and is also in abundance in the cytoplasm of mammalian cells [12]; hence, it is not toxic to the body system. Recently, it was revealed that Cu(I)'s bactericidal mechanisms entail macrophage-derived antibacterial activity. These mechanisms include decreasing osmotic pressure in bacteria by Cu(I), which weakens the bacteria cell membrane. Cu(I) in this case acts as the chemical hydrolase of the sugar–phosphate bond, hydrolysing the bacteria's DNA and RNA, among others [13]. Recently, we reported the antibacterial and antioxidant studies of Cu(I) complexes derived from unsymmetrical formamidine dithiocarbamates alongside triphenyl phosphine ($PPh_3$) [14]. Herein, we are using symmetrical formamidine dithiocarbamate to see how the electronic properties of the complexes would influence the antibacterial and antioxidant potential of the complexes. We also predicted their pharmacokinetics and pharmacological properties using an appropriate in silico method.

## 2. Materials and Methods

The chloroform, acetonitrile, and methanol were of ACS reagent grades; they were purchased from Sigma-Aldrich (Johannesburg, South Africa), and were used without further purification. The reagents 2,6-di(propan-2-yl)aniline (97%), 2,6-Xylidine (99%), Mesitylamine (98%), triphenylphosphane (99%), 1,1,1-Triethoxymethane (99%) and carbon disulfide were also purchased from Sigma Aldrich, while copper(II) nitrate trihydrate (99.5%) and potassium hydroxide (85%) were obtained from Promark Chemicals, Johannesburg, South Africa.

The melting points of complexes **1–3** were recorded using the Electrothermal 9100 (Goteburg, Sweden). The emission spectra were recorded using a PerkinElmer LS 55 fluorescence spectrometer, while the electronic absorption spectra were recorded using the Shimadzu UV-Vis-NIR spectrophotometer (Kyoto, Japan) in the range of 800–200 nm. The IR spectra were recorded on a PerkinElmer Universal ATR spectrum 100 FT-IR (Shelton, CT, USA) in the range of 4000–300 $cm^{-1}$. The $^{13}C$ and $^1H$ NMR spectra were recorded at room temperature on a Bruker Avance$^{III}$ 400 MHz spectrometer (Karlsruhe, German). The elemental analyses and mass spectra of the complexes were recorded on a Vario elemental EL cube CHNS analyzer and the Water synaptic GR electrospray positive spectrometer, respectively.

## 3. General Synthesis Method

### 3.1. Synthesis of Heteroleptic Coper(I) Dithiocarbamate-PPh3 Complexes

The potassium salts of the formamidine dithiocarbamates (**L1**–**L3**) have been communicated elsewhere [5,15], while the metal salt [$Cu(PPh_3)_2NO_3$] was synthesized following a procedure reported by Gysling et al. [16]. We followed the general synthetic technique as reported in the literature [17] to prepare the complexes. Briefly, 1 mmol of the respective potassium dithiocarbamate salts was dissolved in 20 mL acetonitrile in a round bottom flask. To the resultant solution, 1 mmol $Cu(PPh_3)_2NO_3$ dissolved in 10 mL dichloromethane was added drop-wise and stirred for 30 min at room temperature. The resultant yellow solids were collected by filtration, washed three times with ethanol, and then washed twice with ether. The pure products were dried in the oven at 40 °C and stored in a desiccator.

### 3.1.1. Synthesis of [$Cu(PPh_3)_2L1$] (**1**)

The reaction of **L1** (0.37 g, 1 mmol) and $Cu(PPh_3)_2NO_3$ (0.65 g, 1 mmol) in acetonitrile furnished complex **1** as a yellow powder. The yield was 80%. The melting point was 244–245 °C. $^1H$ NMR ($CDCl_3$, 400 MHz): δ (ppm) 2.12 (s, 6H, $CH_3$-Ar), 2.18 (s, 6H, $CH_3$-

Ar), 6.83 (t, 1H, $J_{HH}$ = 7.48 Hz, Ar-H), 6.95 (d, 2H, $J_{HH}$ = 7.48 Hz, Ar-H), 7.11 (d, 2H, $J_{HH}$ = 7.44 Hz, Ar-H), 7.17 (t, 13H, $J_{HH}$ = 7.40 Hz, PPh$_3$), 7.30 (t, 18H, $J_{HH}$ = 7.92 Hz, PPh$_3$), 9.56 (s, 1H, -CH=N). $^{13}$C NMR (CDCl$_3$, 100 MHz) δ (ppm) 17.91, 18.89, 123.10, 127.89, 128.12, 128.22, 129.43, 133.69, 135.82, 148.65 149.59. 215.90. $^{31}$P NMR (121.50 MHZ, CDCl$_3$): δ = 0.0027. IR υ (cm$^{-1}$): 3042(w), 1643(s), 1476(s), 1092(s), 880(s), 492. ESI-TOF MS: $m/z$ (%); [M + 3Na − 5H]$^+$ 979.21, [M − DL1]$^+$ 587.15. UV-Vis (CHCl$_3$, λ$_{max}$, nm): 272 and 330. The analysis calculated for C$_{54}$CuH$_{50}$N$_2$P$_2$S$_2$ was C, 70.76; H, 5.50; N, 3.06; S, 7.00. We found C, 69.71; H, 5.32; N, 2.89; S, 7.01.

### 3.1.2. Synthesis of [Cu(PPh$_3$)$_2$L2] (**2**)

The reaction of **L2** (0.48 g, 1 mmol) and Cu(PPh$_3$)$_2$NO$_3$ (0.65 g, 1 mmol) in acetonitrile furnished complex **2** as a yellow powder. The yield was 74%. The melting point was 218–220 °C. $^1$H NMR (CDCl$_3$, 400 MHz): δ (ppm) 0.98 (d, 6H, $J_{HH}$ = 6.80 Hz CH$_3$-CH), 1.19 (d, 12H, $J_{HH}$ = 6.80 Hz, CH$_3$-CH), 1.23 (d, 6H, $J_{HH}$ = 6.76 Hz CH$_3$-CH), 2.85 (m, 2H, $J_{HH}$ = 6.72 Hz CH-CH$_3$), 3.07 (m, 2H, $J_{HH}$ = 6.64 Hz CH-CH$_3$), 7.01 (s, 1H, Ar-H), 7.07 (t, 3H, $J_{HH}$ = 6.60 Hz, Ar-H), 7.16 (t, 12H, $J_{HH}$ = 7.48 Hz, PPh$_3$), 7.21 (s, 1H, PPh$_3$), 7.29 (d, 18H, $J_{HH}$ = 7.00 Hz, PPh$_3$), 7.36 (t, 1H, $J_{HH}$ = 7.72 Hz, Ar-H), 9.77 (s, 1H, -CH=N). $^{13}$C NMR (CDCl$_3$, 100 MHz) δ (ppm): 10.98, 14.07, 23.00, 23.77, 24.15, 24.26, 24.97, 27.43, 28.97, 30.39, 30.93, 38.76, 122.85, 123.67, 124.04, 128.31, 128.82, 129.04, 129.36, 130.91, 133.70, 134.18, 139.33, 146.04, 146.49, 151.12, and 217.47. $^{31}$P NMR (121.50 MHZ, CDCl$_3$): δ = −0.8479 IR υ (cm$^{-1}$): 3042(w), 1643(s), 1476(s), 1092(s), 880(s), 492. ESI-TOF MS: $m/z$ (%); [M + 3Na]$^+$ 1099, [M − DL2]$^+$ 587.09. UV-Vis (CHCl$_3$, λ$_{max}$, nm): 274 and 333. The analysis calculated for C$_{62}$CuH$_{66}$N$_2$P$_2$S$_2$ was C, 72.38; H, 6.47; N, 2.72; S, 6.23. We found C, 72.18; H, 6.34; N, 2.63; S, 6.30.

### 3.1.3. Synthesis of [Cu(PPh$_3$)$_2$L3] (**3**)

The reaction of **L3** (0.39 g, 1 mmol) and Cu(PPh$_3$)$_2$NO$_3$ (0.65 g, 1 mmol) in acetonitrile furnished complex **3** as a yellow powder. The yield was 75%. The melting point was 214–216 °C. $^1$H NMR (CDCl$_3$, 400 MHz): δ (ppm) 2.09 (s, 6H, CH$_3$-Ar), 2.14 (d, 6H, CH$_3$-Ar), 2.20 (s, 3H, CH$_3$-CH), 2.27 (s, 3H, CH$_3$-Ar), 6.76 (s, 2H, Ar-H), 6.93 (s, 2H, Ar-H), 7.17 (t, 12H, $J_{HH}$ = 7.40 Hz, PPh$_3$), 7.29 (s, 2H, PPh$_3$), 7.31 (d, 16H, $J_{HH}$ = 8.28 Hz, PPh$_3$), 9.53 (s, 1H, -CH=N). $^{13}$C NMR (CDCl$_3$, 100 MHz) δ (ppm): 17.83, 18.84, 20.73, 21.33, 128.04, 128.35, 128.54, 129.78, 129.09, 129.21, 129.44, 129.76, 132.16, 133.73, 134.08, 135.38, 135.89, 137.81, 146.29, 150.17 and 216.05. $^{31}$P NMR (121.50 MHZ, CDCl$_3$): δ = 0.1251 IR υ (cm$^{-1}$): 3055(w), 1633(s), 1478(s), 1024(s), 844(s), 494. ESI-TOF MS: $m/z$ (%); [M − 3CH$_3$]$^+$ 900.08. UV-Vis (CHCl$_3$, λ$_{max}$, nm): 274 and 333. The analysis calculated for C$_{56}$CuH$_{54}$N$_2$P$_2$S$_2$ was C, 71.20; H, 5.76; N, 2.97; S, 6.79. We found C, 70.75; H, 5.60; N, 2.79; S, 6.94.

### 3.2. Single-Crystal X-ray Crystallography

Appropriate single crystals of **1**–**3** were grown by the slow evaporation of a 1,1-dichloromethane/methanol (3:1, $v/v$) solution of each complex. For all of the complexes, single-crystal evaluation and data collection were carried out using a Bruker Smart APEXII diffractometer with Molybdenum Kα radiation (I = 0.71073 Å) equipped with an Oxford Cryostream low-temperature apparatus operating at 100 Kelvin. At different starting angles, reflections were collected and the *APEXII* program suite was utilized to index the reflections [18]. *SAINT* software (Saint Nazianz, WI, USA) [19] was used to execute data reduction, while the *SADABS* multi-scan technique [20] was used to perform scaling and absorption corrections. The *SHELXS* program was used to solve the structures of the complexes by exploring the direct method, while the *SHELXL* program was used for refining [21]. Mercury software [22] was used to draw the structure of each complex. Non-hydrogen atoms were first refined isotropically and then by anisotropic refinement with the full-matrix least square method based on $F^2$ using *SHELXL*. The carbon atom of the dichloromethane solvent in **3** was disordered over an inversion center with 50% site occupancy, and was modelled utilizing PART-1 instruction with a fixed site occupancy

factor of 0.5. The structure refinement parameters and crystallographic data for the available structures are presented in Table 1.

**Table 1.** X-ray crystal data collection and structure refinement parameters for **1**–**3**.

| | **1** | **2** | **3** |
|---|---|---|---|
| Empirical formula | $C_{54}H_{49}CuN_2P_2S_2$ | $C_{62.68}H_{65.36}Cl_{10.60}CuN_2P_2S_2$ | $C_{115}H_{116}Cl_2Cu_2N_4P_4S_4O_2$ |
| Formula weight | 915.55 | 1057.56 | 2036.24 |
| Crystal system | monoclinic | triclinic | triclinic |
| Space group | *P 21/n* | *P-1* | *P-1* |
| $a/\text{Å}$ | 22.4044(4) | 11.5681(4) | 12.5065(3) |
| $b/\text{Å}$ | 9.5916(2) | 15.9024(5) | 13.1610(2) |
| $c/\text{Å}$ | 22.9024(4) | 17.1608(6) | 15.9918(2) |
| $\alpha/°$ | 90° | 104.286(2) | 85.803(10) |
| $\beta/°$ | 109.010(10)° | 104.050(2) | 89.170(2) |
| $\gamma/°$ | 90° | 105.939(2) | 77.974(3) |
| Volume/$\text{Å}^3$ | 4653.17(15) | 2773.51(17) | 2567.55(3) |
| Z | 4 | 2 | 1 |
| $\rho_{calc}\text{g/cm}^3$ | 1.307 | 1.266 | 1.317 |
| $\mu/\text{mm}^{-1}$ | 0.667 | 0.596 | 0.663 |
| F(000) | 1912 | 1113 | 1066 |
| Crystal size/$\text{mm}^3$ | $0.370 \times 0.230 \times 0.140$ | $0.36 \times 0.24 \times 0.14$ | $0.4 \times 0.27 \times 0.14$ |
| 2Θ range for data collection (°) | 1.104 to 28.382 | 1.942 to 28.540 | 1.665 to 28.342 |
| Index ranges | $-29 \leq h \leq 29$ | $-15 \leq h \leq 15$ | $-16 \leq h \leq 16$ |
| | $-12 \leq k \leq 12$ | $-18 \leq k \leq 21$ | $-17 \leq k \leq 17$ |
| | $-30 \leq l \leq 30$ | $-22 \leq l \leq 22$ | $-21 \leq l \leq 21$ |
| Reflections collected | 58617 | 43604 | 9988 |
| Independent reflections | 11,622 [$R_{int}$ = 0.0255] | 13,740 [$R_{int}$ = 0.0179] | 12,357 [$R_{int}$ = 0.0151] |
| Data/restraints/parameters | 11,622/0/550 | 13,740/0/622 | 12,357/1/621 |
| Goodness-of-fit on $F^2$ | 1.041 | 1.018 | 1.022 |
| Final R indexes [I $\geq$ 2σ (I)] | 0.0537, 0.1076 | 0.0316, 0.0756 | 0.0436, 0.0814 |
| Final R indexes [all data] | 0.0537, 0.1166 | 0.0397, 0.0801 | 0.0336, 0.0864 |
| Largest diff. peak and hole (e $\text{Å}^{-3}$) | 0.687 and −0.614 | 0.519 and −0.378 | 0.97 and −1.10 |

### 3.3. In Vitro Antimicrobial Studies

The antimicrobial studies of complexes **1**–**3** were performed using Clinical and Laboratory Standard Institute (CSLI) guidelines [23], with slight modification. They were screened against two Gram-positive bacteria—*Staphylococcus aureus* ATCC 700,699 (methicillin-resistant) and *Staphylococcus aureus* ATCC 25,923—and four Gram-negative bacteria: *Salmonella typhimurium* ATCC 14,026, *Pseudomonas aeruginosa* ATCC 27,853, *Escherichia coli* ATCC 25,922 and *Klebsiella pneumoniae* ATCC 31,488. Ciprofloxacin was used as a reference antibiotic for assessment. The dimethyl sulfoxide (DMSO) showed no antibacterial activity against any of the bacterial strains used for this study at the concentrations tested. The test samples were prepared in dimethyl sulfoxide with a concentration of approximately 1000 μM/mL. The bacteria were inoculated onto nutrient agar (NA) Biolab (Cape-Town South Africa) plates using the streak plate technique and incubated at 37 °C for 18 h. A single colony was isolated and inoculated into 10 mL sterile nutrient broth (NB) Biolab (Cape-Town, South Africa). This was incubated at 37 °C for 18 h in a shaking incubator (100 rpm). The concentration of each bacterial strain was adjusted with sterile distilled water to achieve a final concentration equivalent to 0.5 Mc Farland's Standard (i.e., $1.0 \times 10^8$ cfu/mL) using a densitometer (Mc Farland Latvia). Thereafter, the MHA plates were lawn inoculated with the diluted bacteria using a sterile throat swab. In total, 5 μL of each sample was spotted onto the MHA plates, and the plates were incubated at 37 °C for 18 h and then assessed for antibacterial activity, which was denoted by a clear zone at the point of spotting. Samples that showed antimicrobial potential during the antibacterial screening were tested further to determine their MIC. The samples were serially diluted 10 times to achieve concentrations ranging from 1000 μM/mL to 0.2 μM/mL. In total, 5 μL of each sample at

different concentrations was spotted onto the MHA plates, and the plates were incubated at 37 °C for 18 h and then assessed for their MIC. This was performed in triplicate to ensure reproducibility, and the MIC was determined as the lowest concentration of the compounds at which no visible bacterial growth was observed after incubation.

### 3.4. Determination of Free Radical Scavenging Activity

A 1,1-diphenyl-2-picrylhydrazyl (DPPH) assay was used to evaluate the antioxidant potential of complexes **1**–**3**, and the experiment was carried out as reported by Liyana-Pathirana and Shahidi [24], with slight alterations. To 100 μL 0.1 mM ethanolic solution of DPPH was added 100 μL of the test samples of different concentrations (1.0 mM, 0.75 mM, 0.50 mM, and 0.25 mM). The resulting mixture was vortexed cautiously and left in the dark at 25 °C for half an hour. After half an hour of incubation at 25 °C, the DPPH reduction was measured by reading the absorbance at 517 nm. Different concentrations of ascorbic acid (1.0 mM, 0.75 mM, 0.50 mM, and 0.25 mM) were used as the standard. Equation (1), below, was used to calculate the ability of the complexes to scavenge DPPH radicals.

$$\% \ Scavenging \ Activity = \frac{Absorbance \ control - Absorbance \ of \ sample \times 100}{Absorbance \ control} \tag{1}$$

## 4. Results and Discussion

### 4.1. Synthesis of Cu(I) Complexes **1**–**3**

The protocol for the preparation of **L1**–**L3** has been reported in the literature [5,15], while we followed the procedure reported by Rajput et al. [17] for the synthesis of Cu(I) complexes **1** to **3**. Briefly, an equimolar ratio of each of the respective formamidines, the potassium hydroxide, and the carbon disulfide were reacted under suitable experimental conditions, as previously reported [5], to afford the dithiocarbamate ligands **L1**–**L3**. The heteroleptic Cu(I) dithiocarbamate-PPh$_3$ complexes (**1**–**3**) were synthesized by reacting an acetonitrile potassium salt solution of the ligand with a solution of the metal precursor [Cu(PPh$_3$)$_2$]NO$_3$] in dichloromethane of in equimolar ratios at 25 °C (Scheme 1). The pale-yellow complexes **1** to **3** were obtained in good yields of between 74 and 80% as air stable solids with melting points between 214 and 245 °C.

**Scheme 1.** Synthesis pathway for heteroleptic Cu(I) complexes [Cu(PPh$_3$)$_2$Ln] (n = **1**–**3**).

### 4.2. Spectroscopic Studies

#### 4.2.1. Nuclear Magnetic Resonance

The $^1$H, $^{13}$C and $^{31}$P NMR data of diamagnetic complexes **1**–**3** were all obtained in chloroform. The azomethine proton (NC(*H*)=N) gives an insight into the successful synthesis of **1**–**3** from their free ligands due to an upfield shift from 9.86–10.15 ppm in the

spectra of **L1–L3** to 8.82–9.20 ppm in the spectra of **1–3,** confirming complexation (Table 2) (see Supplementary Materials Figures S1–S6). Generally, there were downfield shifts in the signal of aliphatic protons between the potassium dithiocarbamate salts and the complexes. For example, the peaks for the $CH_3$ protons (*CH*$_3$-Ar) in **L1** at 1.90 and 1.99 ppm shifted to 2.12 and 2.18 ppm in the spectrum of complex **1** (see Supplementary Materials Figures S1 and S4). This noticeable downfield shift can be ascribed to the movement of electron density towards the positive Cu(I) centre in the heteroleptic complexes [25,26]. In addition, there was also a shift of the protons of the aromatic ring of the triphenylphosphine units and the ligands compared to those of the complexes which were observed between 6.83 and 7.36 ppm in the spectra of the complexes. In the $^{13}$C-NMR spectra, a noticeable upfield shift (δ = 1.72–3.47 ppm) in the resonances for the carbon atom of -NCS$_2$ in **1–3** relatively to that of **L1–L3** substantiates the metal-to-sulfur bonding in the complexes (see Supplementary Material Figures S7–S15). A sharp singlet peak appeared at −3.88 ppm in the $^{31}$P-NMR spectrum of the Cu(I) precursor, [Cu(PPh$_3$)$_2$NO$_3$]; upon coordination with **L1–L3**, it appeared at around −0.85–0.13 ppm (see Supplementary Materials Figures S13–S16), confirming the coordination of the Cu atom in the precursor to the dithiocarbamate ligands [27].

**Table 2.** The comparison of -NCS$_2$ and NC(**H**)=N signals for **L1–L3** and **1–3**, and IR bands of (C—N)$_{str}$ and (C=N$_{str}$) for **L1–L3** and **1–3**.

| Ligands (Complex) | δ (-NCS$_2$) ppm | Δ δ | δ NC(H)=N ppm | Δ δ | υ(C=N) cm$^{-1}$ | Δυ | υ(C—N) cm$^{-1}$ | Δυ |
|---|---|---|---|---|---|---|---|---|
| **L1 (1)** | 217.62 (215.90) | 1.72 | 9.86 (9.56) | 0.30 | 1640 (1643) | 3 | 1467 (1476) | 9 |
| **L2 (2)** | 220.94 (217.47) | 3.47 | 10.15 (9.77) | 0.38 | 1639 (1643) | 4 | 1452 (1476) | 24 |
| **L3 (3)** | 218.95 (216.05) | 2.90 | 9.92 (9.53) | 0.39 | 1629 (1633) | 4 | 1477 (1478) | 1 |

### 4.2.2. Fourier Transform Infrared Spectroscopy

The IR spectra of complexes **1–3** exhibited vibrational bands corresponding to the υ(C—NCS$_2$), υ(C—S), and M—S bonds, which are diagnostic bands of dithiocarbamate salts' coordination [28] and υ(C=Nstr) of the azomethine [29]. As seen in Table 2, the υ(C—N$_{str}$) band appeared at higher wave numbers between 1476 and 1478 cm$^{-1}$ relative to those of **L1–L3** (1452–1477 cm$^{-1}$). This shift can be attributed to the mesomeric drift of electrons from the dithiocarbamate units of **L1–L3** towards the metal center in **1–3** [15]. The noticeable enhancement in the υ(C—N) frequency of **1–3** relative to the potassium salts of the free ligands **L1–L3** is an indication of the dithiocarbamate ligands coordinating in a symmetrical bidentate manner due to the dominance of the canonical form R$_2$N=CS$_2$$^-$, thereby indicating the partial double bond character in the thiouride band. A single band around 1024–1092 cm$^{-1}$ (see Supplementary Materials Figures S17–S22) indicates υ(C—S$_{str}$), and thus alludes to the bidentate coordination mode of the ligands **L1–L3** to the metal center [30]. The vibrational bands for υ(C=Nstr) in **1–3** appeared around 1616–1643 cm$^{-1}$, while the ones for **L1–L3** appeared around 1603–1640 cm$^{-1}$. The bands around 429–445 cm$^{-1}$ and 492–500 cm$^{-1}$ were assigned to Cu—P and Cu—S, respectively.

### 4.2.3. Electronic Absorption and Emission Spectroscopy

The electronic absorption spectra of complexes **1–3** in 1,1-dichloromethane solution are presented in Figure 1. The spectra of ligands **L1–L3** displayed two strong absorption bands at around 289–300 nm and 338–345 nm. The absorption bands of **1–3** have the same features as their parent ligands, with two strong absorption bands. However, these two bands are shifted to shorter wavelengths (blue shift), and appear at around 272–277 nm and 330–336 nm. These bands can be allotted to metal-perturbed π—π* intraligand charged transfer transitions within dithiocarbamate and PPh$_3$ ligands [17].

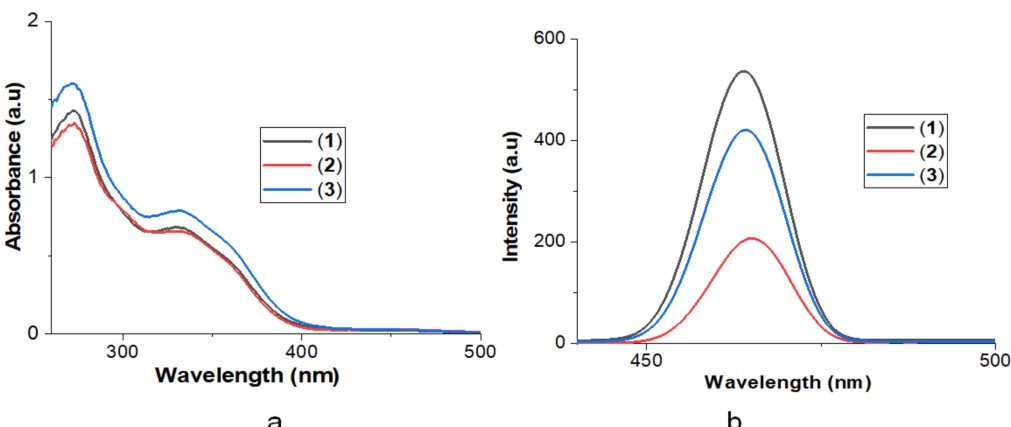

**Figure 1.** (**a**) UV-Visible absorption spectra of **1**–**3** in CH$_2$Cl$_2$. (**b**) Emission spectra of **1**–**3** in CH$_2$Cl$_2$.

When excited at 380 nm in 1,1-dichloromethane solution at 25°C, **1**–**3** displayed an unstructured emission band at about 464 and 465 nm (Figure 1b), emitting a bluish-green light with a large stoke shift, averaging 84–85 nm, which emanated from the admixture of π—π* IL and MLCT in the perturbed coordination environment about the metal atoms [31]. Previous reports have suggested that there is tendency for the complexes to have slight structural changes in the excited state relative to the ground state, due to a small force imposed on them from the bulkiness of the substituents on the dithiocarbamates alongside with the phosphine ligands. For example, a tetrahedral geometry is transformed to a flattened structure, as reported in the case of Cu(I), a d$^{10}$ system, and this influences the luminescent pattern after metal ligand charge transfer transition [32,33]. The calculated energy band gap from the emission spectra of the complexes is 2.673 eV for **1** and **3** and 2.666 eV for **2**.

### 4.3. X-ray Crystal Structures

Appropriate single crystals of **1**–**3** were grown by the slow evaporation of a 1,1-dichloromethane/methanol (3:1, *v*/*v*) solution of each complex. Figure 2 depicts the structures of the complexes, while Table 3 entails the selected bond lengths and angles. The asymmetric units of each of complexes **1**–**3** contain a whole molecule of the Cu(I) dithiocarbamate–PPh$_3$ complex. In the molecular structures of **1**–**3**, the copper atom coordinates to two sulfur atoms of the dithiocarbamate ligand and to two phosphorus atoms of the PPh$_3$ unit. The CuP$_2$S$_2$ core in each of the complexes has a distorted tetrahedral geometry around the Cu(I) center, as can also be observed in the bond angles (Table 2). Using Equation (2), $\tau_4$ for each of the complexes confirmed the proposed tetrahedral geometry [34]:

$$\tau_4 = \frac{360° - (\beta + \alpha)}{141°} \tag{2}$$

where the two largest angles in the four-coordinate species are expressed as α and β. For $\tau_4 = 1$ and $\tau_4 = 0$, the geometries of the complex are perfect tetrahedral and perfect square planar, respectively. The calculated $\tau_4$ values for complexes **1**, **2** and **3** are 0.82, 0.81 and 0.85, respectively, and these fit in Yang et al.'s pseudo-tetrahedral geometry description [34].

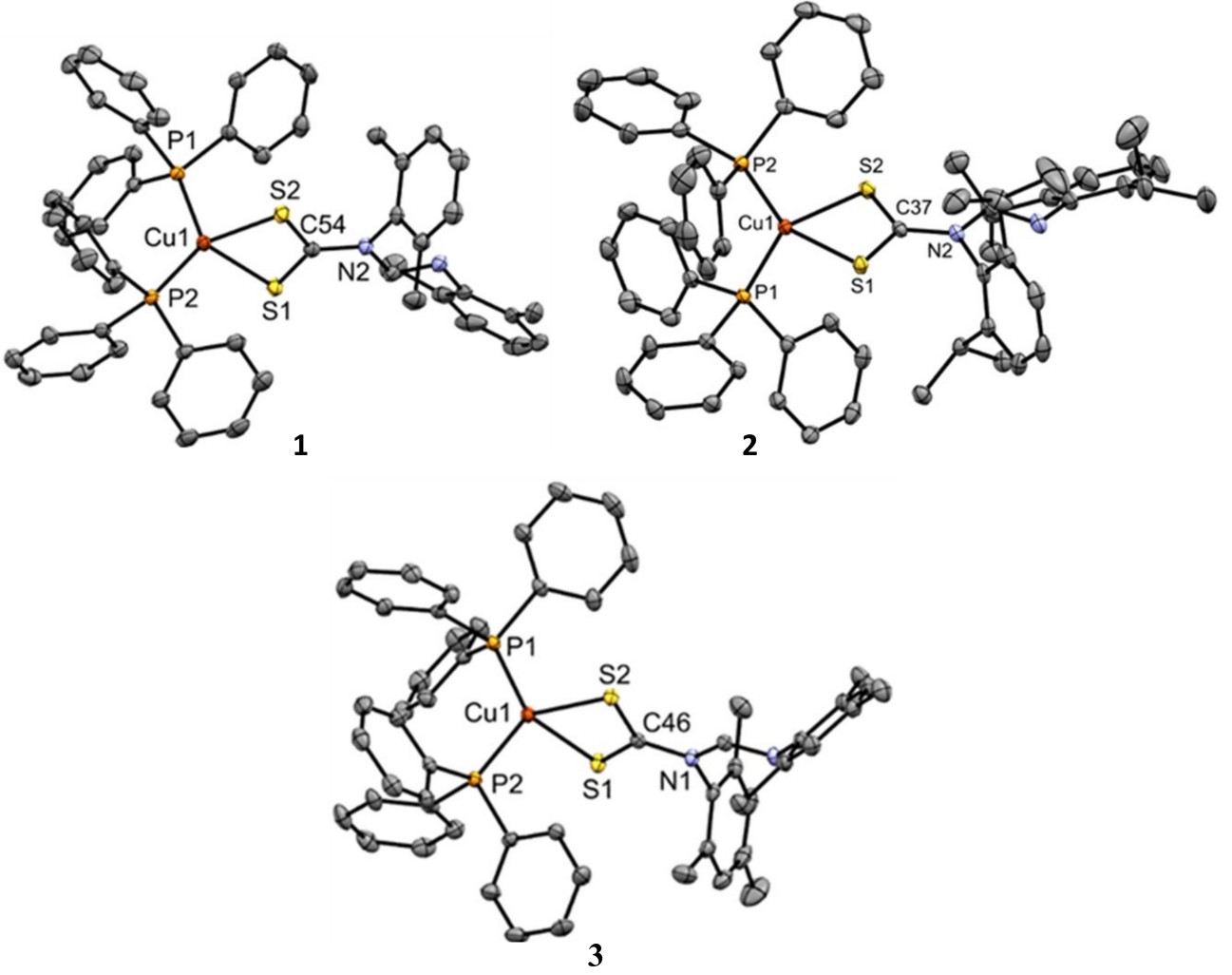

**Figure 2.** Molecular structures of **1**, **2** and **3** drawn at 50% thermal ellipsoid probability. Hydrogen atoms have been omitted for clarity.

**Table 3.** Selected bond length (Å) and angles (°) for complexes **1**, **2** and **3**.

| Parameters | 1 | 2 | 3 |
|---|---|---|---|
| *Bond length* | | | |
| Cu(1)—P(1) | 2.2509(5) | 2.2307(5) | 2.2393(5) |
| Cu(1)—P(2) | 2.2351(7) | 2.2472(7) | 2.2525(5) |
| Cu(1)—S(1) | 2.3891(9) | 2.2472(7) | 2.4060(5) |
| Cu(1)—S(2) | 2.4186(7) | 2.3915(8) | 2.4165(4) |
| $C_{dtc}$—S(1) | 1.696(2) | 1.692(3) | 1.687(2) |
| $C_{dtc}$—S(2) | 1.697(2) | 1.697(3) | 1.704(2) |
| $C_{dtc}$—N | 1.376(3) | 1.384(2) | 1.395(2) |
| *Bond angles* | | | |
| P(1)—Cu(1)—P(2) | 125.73(3) | 126.55(3) | 121.81(2) |
| P(1)—Cu(1)—S(1) | 111.30(3) | 112.60(2) | 115.55(2) |
| P(1)—Cu(1)—S(2) | 105.10(2) | 115.71(3) | 115.17(2) |
| P(2)—Cu(1)—S(1) | 112.74(3) | 107.92(2) | 105.14(2) |
| P(2)—Cu(1)—S(2) | 115.65(2) | 108.10(2) | 114.51(2) |
| Cu(1)—S(1)—$C_{dtc}$ | 83.09(8) | 82.70(5) | 83.34(6) |
| Cu(1)—S(2)—$C_{dtc}$ | 82.17(17) | 83.87(5) | 82.67(6) |
| S(1)—$C_{dtc}$—S(2) | 119.0(1) | 118.72(9) | 119.1(1) |
| S(1)—Cu(1)—S(2) | 75.03(2) | 74.62(1) | 74.61(2) |

The deviation from the perfect tetrahedral arises mainly as a result of the constraint imposed by $CS_2Cu(I)$ chelate rings, and this leads to small bite angles S(1)–Cu–S(2) of 75.03(2)°, 74.62(1)°, and 74.61(2)° for **1**, **2**, and **3**, respectively [12]. Furthermore, the angle formed by the least-square planes of the four-member metallacycles Cu–S–S–C and P–Cu–P for complexes **1**, **2** and **3** are almost perpendicular to each other (87.0°, 87.6° and 87.2°). The $CS_2Cu$ metallacycle in **1**, **2** and **3** deviate from planarity with RMS values of 0.0530, 0.0171 and 0.0404, respectively.

Cu–P and Cu–S are comparable to those found in analogous Cu(1) complexes [35]. The values for the C–N bond in each complex, as shown in Table 3, are in between those published for C=N (1.28 Å) and C–N (1.47 Å) [36]. This indicates the delocalization of π-electrons over the entire $S_2CN$ fragment in the complexes, bringing about the partial double-bond character of the C–N bond [37,38].

### 4.4. Antimicrobial Activities Evaluation

Complexes **1**–**3** were assessed for their antibacterial potential against six strains, *K. pneumoniae*, *P. aeruginosa*, *S. typhimurium*, *E. coli*, *S. aureus* (methicillin resistant) (MSRA) and *S. aureus*, with ciprofloxacin as the standard against **1** to **3**. The antibacterial potential was evaluated using the minimum inhibition concentration (MIC) values (Table 4). It was observed that the complexes exhibited poor-to-moderate antimicrobial activities against the Gram-negative bacteria strains *E. coli*, *K. pneumoniae*, *P. aeruginosa* and *S. typhimurium*, and none were active against the Gram-positive bacteria strains *S. aureus* and MRSA. This difference in the activity of all of the complexes against the two types of bacteria might be due to the differing nature of their cell membranes, which determines the degree to which compounds can penetrate the bacteria [39], with the Gram-positive bacteria having thick layers of peptidoglycan in their cell walls. It is also possible that complexes **1** to **3** were destroyed or modified as they entered the cell walls of the *S. aureus* and the methicillin-resistant *S. aureus* [40].

**Table 4.** MIC of the metal complexes **1**–**3** (μM/mL).

| Complexes | Gram (−) Bacteria | | | | Gram (+) Bacteria | |
| --- | --- | --- | --- | --- | --- | --- |
| | *E. coli* | *S. typhimurium* | *P. aeruginosa* | *K. pneumoniae* | *S. aureus* | MRSA |
| **1** | 3.409 | 13.637 | >1000 | >1000 | N | N |
| **2** | 97.196 | 24.299 | 48.598 | 97.196 | N | N |
| **3** | 26.464 | 105.855 | >1000 | >1000 | N | N |
| Ciprofloxacin [a] | 0.603 | 1.207 | 2.414 | 4.828 | 75.450 | 75.450 |

N = no activity; [a] = standard.

Complex **1** showed fairly good activity against *E. coli* and *S. typhimurium*, while others exhibited moderate-to-low activity against the bacterial strains. Against *P aeruginosa*, complex **2** showed moderate activity, with an MIC value of 48.598 μM/mL, whilst **1** and **3** were only effectively inactive (MIC >1000 μM/mL (highest concentration)). Against *K. pneumonia*, **1** and **3** were active only at >1000 μg/mL, while **2** was active at 97.196 μM/mL (Figure 3). Comparing the antimicrobial potential of the reported compounds with Ciprofloxacin as the reference drug, none of them performed better. We could say it categorically that the replacement of the electron-withdrawing group (chlorine) with the electron-donating group (i.e., methyl, diisopropyl) in the dithiocarbamate of unsymmetrical $N$, $N'$-diaryformamidine dithiocarbamate-PPh$_3$ Cu(I) complexes we previously reported [14] does not influence their antimicrobial potential.

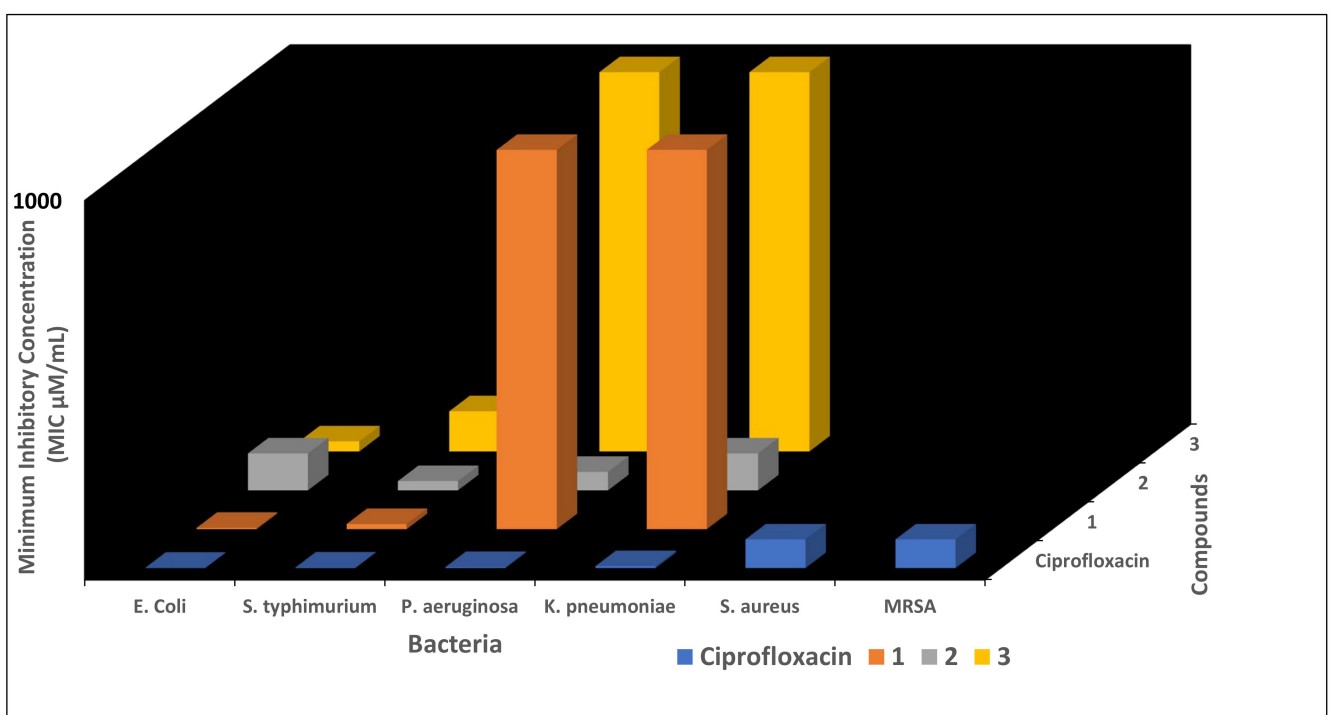

**Figure 3.** MIC values of complexes **1**–**3** against bacteria. The blank spaces represent no activity.

*4.5. Antioxidant Studies*

DPPH Radical Scavenging Ability

The abilities of **1**–**3** to scavenge DPPH radicals are summarized in Table 5; therein, complex **2** has the lowest IC$_{50}$ value of $4.99 \times 10^{-3}$ mM, followed by **3** and **1**. Thus, the antioxidant potential is in the order of **2 > 3 > 1**. It was observed that the ligands' associated electronic factor seems to influence the antioxidant potential of the complexes. Complex **2** with diisopropyl substituents has higher antioxidant ability when compared to **1** with the methyl group. This might be due to complex **2** having more electron density in its aromatic ring as a result of diisopropyl moiety (a better electron-donating group), thereby donating electrons better than complex **1** with the methyl group [29]. Therefore, we could ascribe their free radical ability to their tendency to release the loosely held electrons at the metal center to checkmate the propagation of the free radicals of DPPH. Contrary to the antibacterial studies, the introduction of more electron-donating substituents to the dithiocarbamate ligand backbone of the complexes enhances the antioxidant potential of the previously reported Cu(I) complexes [14]. Generally, the antioxidant activity of **1**–**3** increases as the concentration increases—see Figure 4.

**Table 5.** Antioxidant potential of tested compounds **1**–**3** at varying concentrations using a DPPH assay.

| Complexes | IC$_{50}$ (mM) |
|:---:|:---:|
| **1** | $6.29 \times 10^{-2}$ |
| **2** | $4.99 \times 10^{-3}$ |
| **3** | $5.66 \times 10^{-3}$ |
| Ascorbic acid | $1.04 \times 10^{-3}$ |

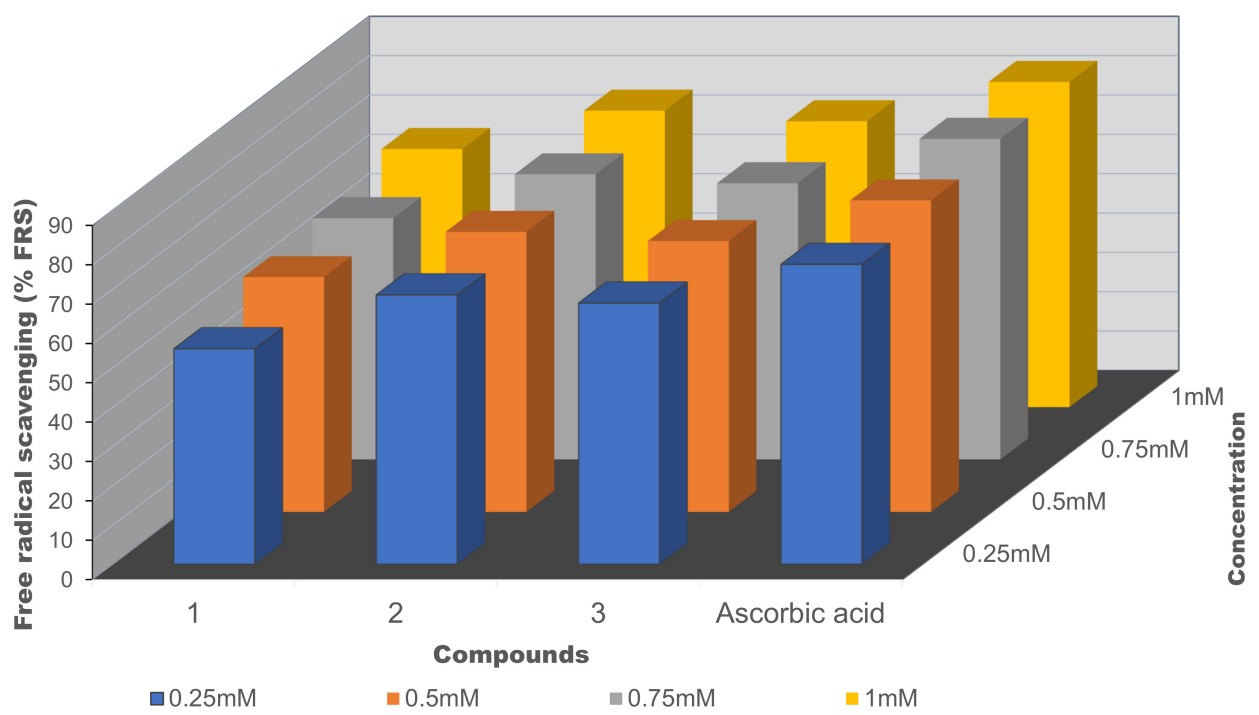

**Figure 4.** Percentage of free radical scavenging against the concentration (mM) of **1**–**3**.

The results presented here are the mean values from three independent experiments.

*4.6. Analysis of the Drug-Likeness and Pharmacokinetics of the Cu Complexes*

The pharmacology and pharmacokinetic properties of complexes **1**–**3** were predicted to pre-determine their drug-likeness, thereby revealing their drug bioactivities. This was achieved using the SwissADME, a web-based analytical tool that computes the physicochemical descriptors, ADME parameters, drug-like nature, and pharmacokinetic properties of molecules [41]. The physicochemical and pharmacokinetics properties of compounds **1**–**3** are presented in Table 6. The estimated values were compared with the accepted standard Lipinski's Ro5 to pinpoint their violations, as potential drug molecules should have relatively slight violations [42,43].

**Table 6.** Predicted physicochemical and pharmacokinetic properties of complex **1**–**6**.

| Physicochemical Properties | 1 | 2 | 3 | Acceptable Threshold (Ro5) |
|---|---|---|---|---|
| Molecular weight (Da) | 916.61 | 1028.82 | 944.66 | <500 Da |
| Log$P$ | 9.83 | 11.87 | 10.38 | <5 |
| LogS (mol/L) | −14.97 | −17.26 | −15.59 | $0 \rightarrow -6$ |
| TPSA (A$^2$) | 93.38 | 93.38 | 93.38 | ≤140 |
| HBA | 1 | 1 | 1 | ≤10 |
| HBD | 0 | 0 | 0 | ≤5 |
| Rotatable bonds | 12 | 16 | 12 | <10 |
| **Pharmacokinetics properties** | | | | |
| GI absorption | Low | Low | Low | |
| BBB Permeant | No | No | No | |
| P-gp Substrate | Yes | Yes | Yes | |
| LogKp (skin permeation) | −1.01 | 0.47 | −0.66 | |

We estimated the molecular weight (MW), lipophilicity (log$P$), aqueous solubility (log$S$), hydrogen bond donor (HBD) and acceptor (HBA) ability, topological polar surface area (TPSA), rotatable bonds (RotB), and skin permeation (Log$K_p$). Pharmacokinetic prop-

erties such as the gastrointestinal (GI) absorption, blood–brain barrier (BBB) permeant, and P-glycoprotein (P-gp) substrate were also predicted. The molecular weight of a compound is one of the factors that determine the extent of absorption and cellular uptake in a living system, and this significantly affects its bioavailability and drug-likeness [38]. A low MW enhances the access of a compound to the target biomolecules while also increasing the concentration at the intestinal epithelium surface and aiding better absorption [44]. According to Lipinski's Ro5, the acceptable MW of a compound is $\leq 500$ g/mol. All of the complexes exceeded this threshold, and thus violated this Lipinski rule. The TPSA is another crucial descriptor that describes the bioactivity of compounds in terms of transportation across a lipid bilayer membrane that is closely packed, such as in the blood–brain barrier (BBB) [45] and gastrointestinal tract (GT). Compounds with a low TPSA have a higher tendency to permeate through the cells relative to those with a high TPSA. All of the complexes have a uniform TPSA of 93.38 $A^2$, and this value agrees with Lipinski's Ro5 (TPSA $\leq 140$), which implies that they have a high propensity to permeate the cells, especially if they are further optimized [38]. The estimated values for Log$P$ and Log$S$ showed that all of the compounds violated the acceptable threshold of Lipinski's Ro5 (<5 for Log$P$ and $0 \to -6$ for Log$S$), indicating poor solubility and lipophilicity. However, methods such as nanoencapsulation and structural fragmentation could be employed to improve the bioavailabilities of such drugs [46].

Their RotBs can predict the molecular flexibility of potential drug compounds, and it increases with molecular weights. Rotable bonds are a single bond, not in a ring, bound to a nonterminal heavy atom [47]. The values for the RotBs for all of the complexes violated Lipinski's Ro5 with RotB counts of <10. Hydrogen bond donors were taken as any heteroatom with at least one bonded hydrogen. In contrast, hydrogen bond acceptors are heteroatom without a formal positive charge, excluding pyrrole, sulfur, and halogen, etc., but including the oxygens bonded to them [47]. Compounds with HBD and HBA counts of $\leq 5$ and $\leq 10$ are considered orally active, according to Lipinski's Ro5. Complexes **1–3** have a uniform HBA count of 1, with no HBD count, and these values indicate their potential to be bioavailable and orally active. When developing an oral drug product, intestinal absorption must be sufficient to be successful [48]. All of the complexes exhibited low gastrointestinal absorption and did not have the potential to permeate through the brain–blood barrier. All of the compounds are P-gp substrates, and the literature has it that the activity of P-gp in the intestine may reduce the oral bioavailability of a wide range of drugs [49]. Interestingly, a number of excipients are usually added to pharmaceutical formulations to disrupt intestinal P-gp and thus enhance the permeability of a substrate drug in the intestine [50]. The rate at which the compounds penetrate the skin is in the order of **2 > 3 > 1**

## 5. Conclusions

Three Cu(I) symmetrical *N*,*N*′-diarylformamidine dithiocarbamate-PPh$_3$ complexes were synthesized and characterized using spectroscopy and analytical techniques. The crystal structures of **1–3** showed that the geometry around the CuS$_2$P$_2$ core is distorted tetrahedral, and their calculated $\tau_4$ values showed that they fit into a pseudo-tetrahedral geometry description. At room temperature, complexes **1–3** displayed interesting luminescent in dichloromethane solutions as a result of an admixture of ILCT and MLCT states. The MIC values for all of the complexes showed that they all had moderate-to-low antibacterial activities against Gram (−) bacteria and displayed no activity against the Gram (+) bacteria used in this study. All of the complexes showed good-to-moderate antioxidant activities, with **2** having the highest free radical scavenging ability. The estimated pharmacological properties showed that all of the complexes had some violations of Lipinski's rule Ro5, and could probably relate to the lower activity against the bacterial strains tested.

**Supplementary Materials:** The following supporting information can be downloaded at https://www.mdpi.com/article/10.3390/inorganics10060079/s1. Figure S1: $^1$H-NMR spectrum for **L1**, Figure S2: $^1$H-NMR spectrum for **L2**, Figure S3: $^1$H-NMR spectrum for **L3**, Figure S4: $^1$H-NMR spectrum for **1**, Figure S5: $^1$H-NMR spectrum for **2**, Figure S6: $^1$H-NMR spectrum for **3**, Figure S7: $^{13}$C-NMR spectrum for **L1**, Figure S8: $^{13}$C-NMR spectrum for **L2**, Figure S9: $^{13}$C-NMR spectrum for **L3**, Figure S10: $^{13}$C-NMR spectrum for **1**, Figure S11: $^{13}$C-NMR spectrum for **2**, Figure S12: $^{13}$C-NMR spectrum for **3**, Figure S13: $^{31}$P-NMR spectrum for **1**, Figure S14: $^{31}$P-NMR spectrum for **2**, Figure S15: $^{31}$P-NMR spectrum for **3**, Figure S16: $^{31}$P-NMR spectrum for $[Cu(PPh_3)_2]NO_3]$, Figure S17: IR Spectrum for **L1**, Figure S18: IR Spectrum for **L2**, Figure S19: IR Spectrum for **L3**, Figure S20: IR Spectrum for **1**, Figure S21: IR Spectrum for **2**, Figure S22: IR Spectrum for **3**, Figure S23: Mass spectrum for **1**, Figure S24: Mass spectrum for **2**, Figure S25: Mass spectrum for **3**. CDC 2155723, CCDC 2155724, and CCDC 2155725 contain the supplementary crystallographic data for complexes **1**, **2** and **3**. These data can be obtained free of charge via http://www.ccdc.cam.ac.uk/conts/retrieving.html, 17 May 2022, or from the Cambridge Crystallographic Data Centre, 12 Union Road, Cambridge CB2 1EZ, UK; fax: (+44)1223-336-033; or via e-mail: deposit@ccdc.cam.ac.uk.

**Author Contributions:** Conceptualization, S.D.O. and B.O.; methodology, S.D.O.; software, S.D.O. and B.O.; validation, S.D.O. and B.O.; formal analysis, S.D.O. and B.O.; resources, B.O.; data curation, S.D.O. and B.O.; writing—original draft preparation, S.D.O. and B.O.; writing—review and editing, S.D.O. and B.O.; visualization, S.D.O. and B.O.; supervision, B.O.; project administration, B.O.; funding acquisition, S.D.O. and B.O. All authors have read and agreed to the published version of the manuscript.

**Funding:** The authors acknowledge the College of Agriculture, Science and Engineering, University of Kwazulu-Natal (UKZN), and the National Research Foundation (NRF) South Africa for financial support (Grant number: 119324).

**Institutional Review Board Statement:** Not applicable.

**Informed Consent Statement:** Not applicable.

**Data Availability Statement:** Not applicable.

**Conflicts of Interest:** The authors declare no conflict of interest.

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
