# Peer review of "Pseudo-Tetrahedral Copper(I) Symmetrical Formamidine Dithiocarbamate-Phosphine Complexes: Antibacterial, Antioxidant and Pharmacokinetics Studies"

_inorganics, doi:10.3390/inorganics10060079_

Round 1
Reviewer 1 Report
A basic study that is somewhat limited in scope. The characterization of the complexes was well-performed, but considering that several similar complexes are known, one would expect a more in depth e.g. SAR. The manuscript requires careful editing of English language and style. Also the authors did not pay attention to the document submitted.
For example Figure 1, page 8, line 258, is a copy of scheme 1, not UV-Visible absorption or Emission spectra of 1 – 3.
Other points are:
1. Into, page 2, line 49. Given that the Cu complexes were prepared to be tested as antimicrobial agents, reference 9 is limiting. In order to better contextualize the study, authors can also refer to https://doi.org/10.3390/chemistry2020026 (and the references therein)
2. although mentioned later in section 4.3, it would be appropriate to specify the crystallization conditions also in the experimental section
3. Figure 3, page 11. the 3D style chosen by the authors to represent their data, overshadows the control data. Also, MIC of y axis are not representative of the data reported in table 4. I personally do not find appropriate to indicate e.g. "15 for MIC values of ≥ 25 μg/mL" as the y axis should in principle reflect MICs in μg/mL.
4. Regarding the DPPH Radical scavenging study. Authors should e.g. discuss the data in relation to the antimicrobial activity of the complexes and contextualize the same. i.e. were other Cu complexes investigated as antioxidants in relation to their antimicrobial properties? What do the data suggest in terms of a possible mechanism of action?
Reviewer 2 Report
In this manuscript the authors are reporting three pseudo tetrahedral copper (I) complexes of stoichiometry [Cu(PPh3)2L] with dithiocarbamate formamidine based ligands. They have synthesized them following literature reported procedures and studied their crystal structure, antibacterial, antioxidant and pharmacokinetic behavior. Overall the idea behind the work is good. But there are many places that appear that the manuscript was not constructed well and missing significant control.
1. It is not feeling smooth while reading the manuscript and it could be improved.
2. In general synthesis section, there is a mistake in writing Phosphorus NMR. 13P should be replaced with 31P, Line -101, 115, 127 should be checked. Also check line 221, 222, 223, it is not clear to understand what the authors tried to say.
3. In table 1 there is a mistake while writing 2 theta values. It could be written either Θ or multiply the values by 2 in the next column. Also for compound 2 the empirical formula comes as fraction while for 1 and 3 it is integers. It would be better if the authors put some explanation somewhere for this.
4. I don’t see the UV/Vis spectra or the Emission spectra anywhere in the main text or in the Supplementary Information. It must be there as it is an important experiment to support the inference.
5. Figure 1 and Scheme 1 are the same. Authors should have paid more attention before submitting.
6. The authors mentioned that ‘The diamagnetic nature of the complexes was confirmed from the NMR spectra’ in line 204. It should be explained more here as it is not very clear to understand how.
7. In the mass spectrum the molecular ion peak should be simulated and it is better to show them both with isotropic distribution.
8. The NMR spectra could be interpreted showing all the protons and carbons in both the drawing and spectrum. The aromatic region where many peaks are aggregated in small region can be expanded and shown as inset with assigning each peaks with multiplicity.
Round 2
Reviewer 1 Report
Overall authors satisfactorily answered my points and the manuscript reads better/is improved. Just a minor point regarding Figure 3 that was removed.
My comment was meant to improve the figure, not necessarily to remove it. Authors may reconsider adding the figure, but with proper axis.
Reviewer 2 Report
In this manuscript the authors are reporting synthesis of three pseudo tetrahedral copper (I) complexes of stoichiometry [Cu(PPh3)2L] with dithiocarbamate formamidine based ligands. They have studied their properties spectroscopically through UV/Vis, Emission, Mass, NMR and studied their crystal structure, antibacterial, antioxidant and pharmacokinetic behavior if they have any. The idea behind the work is good but still it has some weakness. But the authors have addressed the points raised after first round of reviewing and tried to modify it to their best and explained why they are not able to do some of the modifications recommended. Considering their effort to modify, this can be accepted.
